# Differences in the Pro/Antioxidative Status and Cellular Stress Response in Elderly Women after 6 Weeks of Exercise Training Supported by 1000 mg of Vitamin C Supplementation

**DOI:** 10.3390/biomedicines10102641

**Published:** 2022-10-20

**Authors:** Małgorzata Żychowska, Ewa Sadowska-Krępa, Elisabetta Damiani, Luca Tiano, Ewa Ziemann, Alicja Nowak-Zaleska, Patrycja Lipińska, Anna Piotrowska, Olga Czerwińska-Ledwig, Wanda Pilch, Jędrzej Antosiewicz

**Affiliations:** 1Department of Biological Foundations of Physical Culture, Kazimierz Wielki University, 85-091 Bydgoszcz, Poland; 2Department of Nature Sciences, Gdansk University of Physical Education and Sport, 80-336 Gdansk, Poland; 3Institute of Sport Sciences, Jerzy Kukuczka Academy of Physical Education, 40-065 Katowice, Poland; 4Department of Life and Environmental Sciences, Marche Polytechnic University, 60121 Ancona, Italy; 5Department of Athletics, Strength and Conditioning, Poznan University of Physical Education, 61-871 Poznan, Poland; 6Institute of Basics Sciences, Faculty of Physiotherapy, University of Physical Education, 31-571 Kraków, Poland; 7Department of Bioenergetics and Exercise Physiology, Medical University of Gdansk, 80-210 Gdansk, Poland

**Keywords:** exercise, genes encoding heat shock proteins, total oxidative status, vitamin C supplementation

## Abstract

Vitamin C supplementation and exercise influence pro/antioxidative status and the cellular stress response. We tested the effects of exercise training for 6 weeks, supported by 1000 mg of vitamin C supplementation in elderly women. Thirty-six women were divided into two groups: a control group (CON) (n = 18, age 69.4 ± 6.4 years, 70.4 ±10.4 kg body mass) and a supplemented group (SUPP) (n = 18, aged 67.7 ± 5.6 years, body mass 71.46 ± 5.39 kg). Blood samples were taken twice (at baseline and 24 h after the whole period of training), in order to determine vitamin C concentration, the total oxidative status/capacity (TOS/TOC), total antioxidant status/capacity (TAS/TAC), and gene expression associated with cellular stress response: encoding heat shock factor (*HSF1*), heat shock protein 70 (*HSPA1A*), heat shock protein 27 (*HSPB1*), and tumor necrosis factor alpha (*TNF-α*). We observed a significant increase in TOS/TOC, TAS/TAC, and prooxidant/antioxidant balance in the SUPP group. There was a significant decrease in *HSPA1A* in the CON group and a different tendency in the expression of *HSF1* and *TNF-α* between groups. In conclusion, vitamin C supplementation enhanced the pro-oxidation in elderly women with a normal plasma vitamin C concentration and influenced minor changes in training adaptation gene expression.

## 1. Introduction

Antioxidant supplementation and its influence on human health have been investigated in relation to diseases such as cardiovascular, cancer, neurodegenerative, and other pathologies associated with oxidative stress [1,2,3,4]. Vitamin C supplementation promotes an increase in antioxidant capacity and in turn protects against oxidative stress and its consequences. There is also evidence that vitamin C supplementation directly reduces the cell damage caused by several stress conditions and protects against the cellular stress response [5]. Frei et al. [6] postulated that daily consumption of 200 mg of vitamin C is optimal to maximize the vitamin’s potential health benefits without adverse effects in adults. However, there is no evidence on significant health benefits arising from daily vitamin C supplementation, for healthy people on a balanced diet and without deficiencies in antioxidant vitamins. Furthermore, supplementation of vitamin C may cause antioxidant and prooxidant actions simultaneously and does not influence the pro-antioxidant balance [7,8,9]. Prooxidative/antioxidative reactions are not only associated with vitamins, which represent non-enzymatic defenses, they also involve intracellular enzymes. Therefore, other factors should be taken into consideration. For example, the degree of obesity may play a role in differences regarding prooxidative/antioxidative protection [10]. Regular physical activity, both professional or recreational, can induce changes in the antioxidant status and cellular protection, which include increases in the antioxidant capacity and decreases in the proinflammatory status, accompanied by increases in the anti-inflammatory response. Modulation of these processes is characterized by an individual’s cellular stress response to the adaptation to exercise [11,12]. However, the effects of antioxidant supplementation during training may depend on the type of strength or aerobic exercise. The literature data for both aerobic and strength training are ambiguous. Bjørnsen et al. [13]; Close et al. [14], and Gomez-Cabrera et al. [15] postulated that high doses of vitamin C may have a negative effect on training adaptation. Moreover, they noted that age did not have any impact on such responses. Vitamin C and E supplementation can also alleviate the difficulty of muscle growth by attenuating pro-inflammatory responses, and in consequence inhibiting hypertrophy [16]. Therefore, long-duration antioxidant supplementation is not recommended for training related adaptation of skeletal muscles [16]. Antioxidant supplementation after a 60-min steady state ride with 70% of maximal oxygen consumption (VO_2_ max.) in trained cyclists resulted in reduced muscle damage but no changes in physical performance [17]. Vitamin E combined with vitamin C supplementation was more effective compared to vitamin C supplementation alone [12]. On the other hand, combined vitamin E and vitamin C supplementation may also lead to impaired training adaptation, independently of the type of training [12]. In particular, vitamin supplementation has been associated with an arrest in muscle mass formation during strength training, and it also impaired mitochondrial biogenesis mediated, by downregulation of genes normally positively influenced during endurance training (such as peroxisome proliferator-activated receptor-gamma coactivator, PGC-1α, [18]). Cliffort et al. [19], in his review, indicated that there are not enough data to conclude that vitamin C alone or combined with vitamin E supplementation impairs adaptations in physiological functions, including at molecular level. Antioxidant supplementation may influence gene expressions that are easily stimulated by changes in the pro/antioxidative status, especially those that are associated with the stress response [5]. Combined vitamin E and vitamin C supplementation ameliorated the cellular stress response caused by cyclic heat stress in rats [20]. However, antioxidant supplementation is not only associated with vitamin C, which when supplemented alone may result in weaker or different effects on the prooxidative/antioxidative balance and on gene expression [20]. Data regarding the impact of vitamin C supplementation on the gene expression involved in cellular stress response in humans subjected to regular exercise are limited. The cellular stress response depends on the activation of signaling pathways, mainly associated with heat shock factor 1 (HSF1) and nuclear factor kappa B (NF-kB) pathways. NF-kB, HSF1, and tumor necrosis factor alpha (TNF-α) play important functions in immune response; they are involved in anti- and pro-apoptotic functions [5], as well as modified cell survival under stressful conditions [21]. Vitamin C’s influence on reactive oxygen species (ROS) production decreased TNF-α production and stress-induced heart damage in mice with a vitamin C deficiency [22]. TNF-α has an affinity for TNF-α receptor 1 and 2 (TNFR1 and TNFR2). Furthermore, the TNF-α cascade of reactions could be initiated by two independent pathways, which lead to two opposite effects: either apoptosis or changes leading to anti-apoptotic effects via the NF-κB dependent pathway [22]. Additionally, activation of the genes encoding *NF-kB* or *HSF1* pathways occurs indirectly, due to an increase in ROS production [23]. Heat shock protein expression is dependent on *HSF1* activation and is easily induced by many stressors, such as oxidative stress, physical effort, and changes in temperature [24,25,26]. Increased mRNA encoding heat shock protein 70 (*HSPA1A*) and heat shock protein 27 (*HSPB1*), induced by oxidative stress through activation of the HSF-1 pathway, displayed an anti-apoptotic action and prevented DNA damage, independently of exercise [27]. However, *HSPB1* overexpression has been mainly associated with the degradation of damaged proteins caused by exercise [26,28]. Recently, we investigated the effects of vitamin C supplementation on genes associated with iron metabolism and inflammation in elderly women; we observed that vitamin C supplementation decreased inflammation and decreased ferritin heavy (*FTH*) and ferritin light (*FTL*) mRNA during adaptation to physical training [7]. In another study, we concluded that slight downregulation in interleukin 6 (*IL6*) and interleukin 10 (*IL10*) mRNA was positively correlated with adaptation to training [9]. Accordingly, we postulated that vitamin C supplementation had a greater effect on gene expression, which may be linked to a lower oxidative stress within cells, rather than a direct influence on the plasma prooxidative/antioxidative balance [7]. The influence on ferritin mRNA levels could indicate a positive effect of supplementation on adaptation to training in elderly women [7]. However, in young figure skaters, vitamin C supplementation decreased the expression of *HSPA1A* and *HSPB1*, which indicated less stress caused by training during the conditioning camp but, at the same time, being more prone to apoptosis [5].

Therefore, because of the discrepancies reported in the literature on the health benefits of vitamin C supplementation during exercise, in the present study, we investigated the changes in the main transcriptional factors involved in the cellular stress response and genes dependent on these factors (encoding heat shock proteins), in elderly women after exercise training and vitamin C supplementation. We hypothesized that vitamin C supplementation would cause a decreased *TNF*-α and *HSF1* expression, and subsequently, a decreased *HSPA1A* and *HSPB1* expression during 6 weeks of training supported by 1000 mg of vitamin C. Confirmation of this hypothesis would allow the conclusion that vitamin C supplementation helps in the molecular adaptation to training in elderly women.

## 2. Materials and Methods

The results presented in this study were derived from four scientific centers: Gdansk University of Physical Education and Sport, Poland; Kazimierz Wielki University in Bydgoszcz, Poland; University of Physical Education in Krakow, Poland; and Marche Polytechnic University, Ancona, Italy.

### 2.1. Participants

Thirty-six elderly women participated in this study. The women were randomized into two groups: a) the control group (CON) (n = 18, age 69.4 ± 6.4 years and 70.4 ±10.4 kg body mass) received placebo during 6 weeks of training; and b) the supplemented group (SUPP) (n = 18, age 67.7 ± 5.6 years and body mass 71.46 ± 5.39 kg) were supplemented orally with vitamin C (1000 mg daily) for 6 weeks of training. The study was double-blinded. Participants’ body mass was analyzed using an InBody 720 device (Inbody, Biospace Co., Ltd., Seoul, Korea).

The main characteristics of both groups are summarized in Table 1. The recruitment criteria were as follows: females aged between 60 and 80 years; medical permission to participate in the trial; no participation in other projects; a low level of physical activity; and proper balanced diet, in terms of energy content and basic nutrients. Only women with an adequate dietary intake of energy, protein, fat, and carbohydrates, according to the guidelines established for the given age group of the Polish population [29], were considered eligible for the project. The exclusion criteria were as follows: the age range of women outside 60–80 years; movement limitations or ongoing injuries; endoprostheses or other conditions limiting the possibility of performing tests; neoplastic diseases; advanced cardio-respiratory diseases; arrhythmias; history of arterial congestion; hypertension (>160/100 mm Hg); transient ischemic attacks; thyroid malfunction; diabetes; smoking; total cholesterol levels >300 mg/dL; on a weight loss diet; and use of anti-inflammatory drugs. None of the women had a disease requiring ongoing treatment, and all avoided supplementation for 3 months prior to the trial.

### 2.2. Diet Analysis

At the time of recruitment, prior to training and supplementation, all participants recorded the number of meals consumed over a 5-day period, i.e., 3 working days and 2 non-working days. Their records were analyzed in the Diet 5.0 software developed by the Food and Nutrition Institute (Warszawa, Poland). Meal sizes were estimated using a photo album of products and dishes [30] to calculate the average daily energy intake and the percentage of proteins, fats, and carbohydrates. Participants were asked to maintain their nutritional habits throughout the duration of the study. To avoid seasonal changes in nutrition, the study spanned November and February (2019/2020; 2020/2021; 2021/2022).

### 2.3. Training

The training schedule was based on a previously applied training program [7,9]. A six-week training period, consisting of moderate-intensity Nordic walking, resistance training, and stabilization training, was designed according to the principles of health-related training. The program was conducted with three weekly sessions of 60 min. The exercise intensity was maintained at a level below 130 beats/min (Polar H10, Finland).

### 2.4. Supplementation

Participants were randomized in a double-blind procedure to the SUPP group (vitamin C, Max VitaC 1000, Colfarm, Mielec, Poland), or CON placebo group (cellulose tablets, Colfarm, Mielec, Poland). Supplements were taken during breakfast for 6 weeks. The choice of vitamin C dose was based on the one most often recommended by doctors and pharmacists based on guidelines for nutritional supplements.

### 2.5. Determination of Maximum Oxygen Uptake

Assessment of maximal oxygen consumption (VO_2_) was performed to establish the individual’s intensity of training. It was determined using a cycloergometer (Ergoline Ergoselect 150p, ergoline GmbH, Bitz, Germany) and a gas analyzer (Jaeger OxyconPro, Chatham, IL, USA). The measurement procedure consisted of 2 min for registration of the participant’s resting values, followed by 5 min of a warm-up with 30 W load and 60 rpm cadence. The proper exercise test involved a gradual increase in load by 10 W every minute. The test was discontinued when a participant was unable to continue or when other symptoms indicated the need to end the trial. The results of the VO_2_ max are presented in Table 1 (Section Results).

### 2.6. RNA Isolation

For this purpose, the procedure of Chomczynski and Sacchi [31], previously described by Żychowska et al. [7,9], was used. Venous blood (2 mL) was collected into vacutainer tubes with ethylenediaminetetraacetic acid (EDTA) as an anticoagulant. To remove erythrocytes, blood samples were treated with red blood cell lysis buffer (RBCL) (A&A Biotechnology, Gdynia, Poland) on ice (20 min) and then centrifuged for 10 min at 3000× *g* at 4 °C. The obtained leukocytes were lysed with Fenozol (A&A Biotechnology, Gdynia, Poland) and incubated for 5 min. After incubation, 200 µL of chloroform (POCH, Gliwice, Poland) was added and the suspension was shaken. The aqueous phase was then transferred to an Eppendorf tube, where 500 µL of isopropanol (POCH, Gliwice, Poland) was added to precipitate RNA. The samples were then centrifuged for 30 min at 10,000× *g*, at 4 °C. The resulting RNA precipitate was washed twice in 1 mL of 75% ethanol and centrifuged for 5 min at 7500× *g*, at 4 °C. The ethanol was then pipetted off and the pellet was left to dry. The dry RNA was diluted in 20 μL of molecular grade water. The purity and quality were assessed spectrophotometrically (BioPhotometer Plus, Eppendorf, Germany).

### 2.7. Reverse Transcription and qRT-PCR

For this purpose, 1000 ng of pure RNA (A260/280 ≥ 1.7) (AffinityScript QPCR cDNA Synthesis Kit: Agilent, Poland) was used in an Eppendorf Mastercycler Gradient 5331. The reaction run profile followed the manufacturer’s instructions. The obtained cDNA was diluted 10-fold immediately before the PCR reaction. Expression of each gene was detected by quantitative real-time PCR reaction (qRT-PCR, Aria, Agilent, Polish Department). The following primers were used to amplify gene expression:

For *TUBB* (NM_001293213): F: CTAGAACCTGGGACCATGGA and R: TGCAGGCAGTCACAGCTCT.

For *HSPA1A* (NM_005345.6) F: ACTCCCGTTGTCCCAAGGCTT C and R: TCTGTCGGC TCCGCTCTGAGA,

For *HSPB1* (NM_001540.5) F: AAGGATGGCGTGGTGGAGATCA and R: GAGGAAACTTGGGTGGGGTCCA,

For *TNF-α* (NM_000594) F: GATCTCTCTCTAATCAGCCC and R: GCAATGATTCCA AAG ACCTGCCC.

### 2.8. Assessment of Plasma Vitamin C Level

Plasma vitamin C concentration was determined by the method of Robitaille and Hoffer [32]. For this purpose, cold trichloroacetic acid (TCA) 20% (0.4 mL) and cold dithiothreitol (DTT) 0.2% (0.4 mL) were added to 0.2 mL of plasma. The samples were then centrifuged for 10 min at 10,000× *g*, at 4 °C. The supernatant was then frozen at −80 °C and stored until further determinations. Determination of vitamin C concentration was performed in an accredited laboratory in Krakow (Poland) by high-performance liquid chromatography (HPLC) with UV detection (wavelength: 245 nm). The chromatography was carried out on a reversed-phase chromatography column (RP-18) (Merck, Darmstadt, Germany) with a length of 25 cm, a diameter of 4.6 mm, and a grain diameter of 5 μm, and the mobile phase pH was 2.7. A UV detector 2140 Rap1d Spectral Detector Optical Unit (LKB Bromma, Sweden) and a Rheodyne^®^ (Model 7010) injector (Rheodyne, Germany) were used. The vitamin C retention time was 3 min [32].

### 2.9. Biochemical Analysis

Venous blood was collected into BD Vacutainer tubes (Becton Dickinson, Franklin Lakes, NJ, USA) and centrifuged for 10 min at 3000× *g*, at 4 °C. The plasma was then stored at −80 °C, until biochemical assays were performed. Total oxidative status/capacity (TOS/TOC) and total antioxidant status/capacity (TAS/TAC) were evaluated using diagnostic kits (Immundiagnostik AG, Germany) for PerOx and ImAnOx, respectively. In the PerOX assay, total lipid peroxides level was measured in the sample. PerOx determination was based on the reaction between lipid peroxides and peroxidase with use of photometric detection. ImAnOx determination was based on the reaction of antioxidants present in the sample with added hydrogen peroxide; the amount of which remaining after the reaction was determined photometrically. Using both indices, we calculated the TOS/TOC to TAS/TAC ratio as the prooxidant/antioxidant balance.

### 2.10. Statistical Analysis

All data were presented as mean values (x) and standard deviations (SD). The data normality distribution was established using a Shapiro–Wilk test. The results obtained before and after 6 weeks of training were compared using parametric tests for normally distributed data, or non-parametric (Wilcoxon) tests for non-normally distributed data. A paired t-test was applied to evaluate within-person changes and differences between groups. The genetic data were collected, and relative gene expressions were analyzed in Microsoft Excel 2015. The level of mRNA was calculated using the comparative method of Schmittgen and Livak [33]. The mRNA levels of the tested genes were described as the differences in the cycle threshold values normalized to TUBB mRNA levels, i.e., ΔCT = CT of gene—CT of TUBB. The data were transformed into linear values, and statistical significance was evaluated using a Shapiro–Wilk test to assess for normal distribution, and a paired or un-paired test was used accordingly, for comparison of results before and after the 6-week training period in both groups. Student’s t-test and two-way ANOVA were used to determine the significance of differences between the groups. All analyses and figures were generated using GraphPad Prism 6.0. The level of significance in all tests were considered statistically significant at *p* < 0.05.

## 3. Results

Body mass, relative VO_2_ max, and vitamin C concentration were observed during 6 weeks of training, with and without 1000 mg vitamin supplementation. All results are presented in Table 1.

At baseline, there were no significant differences between groups in all tested parameters. Body mass decreased in response to the training period; however, a significant shift was only observed in the CON group (*p* < 0.05). VO_2_ max did not change significantly in the reported period; however, 2-way ANOVA showed significant differences for interaction and row factor. This parameter was slightly increased only in the SUPP group. Vitamin C concentration increased significantly in the SUPP group, while it remained almost unchanged in the CON group.

The TOS/TOC at baseline was high in both groups: CON (448.3 ± 122.9) and SUPP (474.8 ± 111.5). Similarly, participants in the two groups also had a high TAS/TAC (340.0 ± 62.1 in CON and 329.6 ± 109.7 in SUPP). There was a significant increase in TOS/TOC in the SUPP group at 6 weeks post supplementation (from 329.6 to 519.2 ± 116.5 mmol/L, *p* = 0.03), and this change was accompanied by an increase in TAS/TAC. The increase in antioxidant status was significant in the SUPP group (from 329.6 ± 109.7 to 393.5 ± 99.0, *p* < 0.0001, Figure 1B), while in the CON group this remained unchanged (Figure 1A,B). After 6 weeks of training, with or without supplementation, the differences between groups in antioxidant status were significant (*p* = 0.04). The applied 2-way ANOVA confirmed differences between groups for TOS/TOC and TAS/TAC (row factor and time *p* < 0.05).

There were significant differences between groups in the TOS/TOC and TAS/TAC balance at baseline. Specifically, we observed that increases in TOS/TOC outperformed increases in TAS/TAC; this was especially visible and significant in the SUPP group (*p* = 0.016).

*HSPA1A* (Figure 2B) was significantly decreased in the CON group after 6 weeks of training (from 0.22 ± 0.14 to 0.14 ± 0.09, *p* = 0.04), while it remained unchanged in the SUPP group. There were no significant differences with the other genes. However, although there were no significant differences in the remaining genes, *HSF-1* showed a tendency to decrease in the CON group, while it remained almost unchanged in the SUPP group (Figure 2A). *HSPB1* was slightly increased in both groups (Figure 2C). The opposite tendency was observed for TNF-α in both groups; there was a clear tendency to decrease in the CON group and a slight tendency to increase in the SUPP group (Figure 2D). However, in the CON group after 6 weeks of training without supplementation, individual variations in the data were high. Similarly, although not significant, the same tendency was noted for *TNF-α* and *HSF-1* in this group. In conclusion, we observed that an increase in the antioxidant capacity of the CON group was accompanied by an increase in the oxidative stress status.

## 4. Discussion

In this study we assumed that vitamin C supplementation would increase the TAS/TAC and pro/antioxidative balance and thus reduce peroxidation. Genes associated with cellular stress response are easily induced by oxidative stress; therefore, during supplementation, the expression of these genes should be reduced. The current literature associated with gene expression and adaptation to exercise at a molecular level are limited. Most data are related to the influence of this supplementation on physiological and biochemical indicators. Thus, some authors postulated no influence on adaptation to training or an impediment in adaptation of skeletal muscle mass (anaerobic exercises) or capability for endurance training (aerobic exercises) [16,18,34]. The obtained data for gene expression in present study did not confirm our hypothesis associated with the expression of *HSPA1A*, *HSPB1*, *TNF-α*, and *HSF-1*. There was a small decrease in the antioxidant capacity after exercise without supplementation in the CON group and a significant increase in the antioxidative capacity in the SUPP group. A decrease in the antioxidant capacity caused by exercise confirms the data reported by Aguilo et al. [35]. Thus, increases in TAS/TAC were dependent on vitamin C concentration in the SUPP group. Unfortunately, this change in TAS/TAC was also accompanied by an increase in TOS/TOC. Finally, the oxidative/antioxidative balance showed a significant increase in the SUPP group and only a slight increase in the CON group. Thus, in elderly women with an appropriate plasma vitamin C concentration, both oxidation and antioxidation actions, resulting from the properties of vitamin C as a chemical compound (gamma lactone of hexanoic acid with the endiol system), were observed. The possibility of prooxidative and antioxidative action was indicated by several authors, who reported a “second face” of vitamin C [36,37]. The chemical mechanism of prooxidative action relies on catalyzation of the reduction of free transition metal ions, which causes the formation of oxygen radicals and the reduced iron ions react with hydrogen peroxide to form reactive hydroxyl radicals or peroxide ions [36]. It was shown that vitamin C could mobilize endogenous copper in human peripheral lymphocytes, leading to oxidative breakdown of DNA [38]. Vitamin C could influence the toxicity of chrome ions (CrVI), both in oxidative and antioxidative directions [37]. Some current studies have indicated that vitamin C is able to modulate gene expression and cellular function [34]. Therefore, we assumed that vitamin C supplementation could have been a reason for the different changes in the genes involved in the cellular stress response between the supplemented and un-supplemented groups. Moreover, the prooxidative and antioxidative action of vitamin C could be also a reason why several authors suggested not supplementing a high dose of vitamin C and/or E during intensive strength training [16]. However, it is well known that vitamin C positively influences vascular function, without improved exercise possibilities [16].

Second, the gene expression for *HSF1*, *HSPB1*, and *TNF-α* had the tendency to increase in the group supported by supplementation after the training period. This suggests that vitamin C supplementation stopped the adaptive changes caused by exercise for some gene expressions, only via increasing the TOS/TAS balance. The significant decrease in *HSPA1A* in the CON group could have been a result of the applied exercise training. In our opinion, the lack of changes in genes easily induced by exercise in the SUPP group (e.g., *HSF1*, *HSPA1A*, *HSPB1*, or *TNF-α*) differently from the CON group could have been a result of the prooxidative action of vitamin C supplementation, in agreement with a significant increase in the TOS/TAS balance.

In the current literature, the data associated with vitamin C supplementation and its influence on antioxidant capacity are contradictory. Some studies indicated that antioxidant supplementation increases TAS/TAC, decreases TOS/TOC, or has a prooxidative action. For instance, Zoppi et al. [39] showed decreases in lipid peroxidation and muscle damage in professional soccer players, while Morton et al. [40] reported that vitamin C only slightly decreased malondialdehyde and oxidized low-level lipoprotein levels. Morton et al. postulated that their results could be dependent on the applied dose; they used a high supplement dose of 20 mg/kg vitamin C in rats [40]. Supplementation with 500 mg vitamin C and 1000 mg vitamin E in elderly women produced no effect on oxidative stress and HSP72 expression, as shown by Simar et al. [41]. Similar data were reported by Cumming et al. [42], after 1000 mg vitamin C and 235 mg vitamin E supplementation in muscle cells. In a previous study on 24 elderly women (12 in the CON group and 12 in the 1000 mg vitamin C SUPP group), who were subjected to the same training regime and for the same period (6 weeks), we observed no changes in the pro/antioxidative balance; however, supplementation increased TAS/TAC accompanied by increased TOS/TOC. According to the literature, the oxidative action of vitamin C supplementation has previously been reported by many authors [7,15,39]. Vitamin C promoted ROS such as H_2_O_2_, hydroxyl radical (^.^OH), superoxide radical anion O_2_^−^, and ferryl ions [7,8]. This effect was reported for doses of vitamin C between 250 and 500 mg/day after 12 weeks of exercise by Bunpo and Anthony [43], which are lower than in our study.

According to our results, we postulated that increases in TOS/TOC in the SUPP group directly influenced the increase in *HSPA1A*, *HSPB1*, *TNF-α*, and *HSF1* mRNA. Unfortunately, there are limited studies considering these genes and the multidirectional action of the proteins encoded by them. In the current literature, there are some data concerning changes in HSP proteins during exercise; however, the gene expression in human leukocytes have not been frequently reported [5]. In a previous study, we showed that vitamin C supplementation decreased the gene expression for *HSPA1A* and *HSPB1* in leukocytes in young figure skaters during a training camp [5]; the present data are not coherent with this. However, there are many differences between these studies, such as the different ages of participants, training load, dose of supplementation, and level of physical activity etc. Additionally, we did not previously determine plasma vitamin C status, and this determination could explain the different outcomes. However, it is possible, that all these factors could have influenced the differences between these studies. Interpretation of changes in HSP expression is difficult. Some data showed a decrease in oxidation, which caused a reduction in HSP synthesis and in turn resulted in lowered stress in the body [35,44]. On the one hand, it can be concluded, that the thermal tolerance was also decreased [40]. Moreover, a decrease in the expression of genes encoding HSP70 or HSP 27 reduced anti-apoptotic protection, which may not be beneficial, especially in the elderly of the CON group. In conclusion, in our study the CON group showed adaptation changes to exercise, while the vitamin C SUPP group prevented these changes, and the anti-apoptotic protection was unchanged in this group.

Petersen and Petersen [45] postulated that decreases in TNF-α as an effect of exercise are associated with IL6 production in muscle cells after exercise, which stimulate anti-inflammatory cytokines such as IL10. Thus, regular exercise suppresses TNF-α, which is important for maintaining insulin sensitivity [45]. In our study, TNF-α tended to decrease only in the CON group. It seems, that vitamin C opposes this change in the SUPP group. The interpretation of these changes is also associated with TOS/TAS. For example, Kim et al. [22] postulated that vitamin C decreased TNF-α and promoted stress-induced damage in the heart, in mice with a vitamin C deficiency.

After the training period, the body mass significantly decreased only in people from the CON group. Johnston et al. [46] reported that plasma vitamin C inversely correlates with body mass. This effect was not observed in our participants. In our opinion, the reduction in inflammation could have influenced fat loss, and it is possible that this was a result of vitamin C supplementation, which we previously investigated [9]. It is well documented that physical exercise influences body mass loss, due to the increased consumption of ATP, and this effect was clearly visible in participants from the CON group. In this experiment, we noted a slight effect of mass loss in the SUPP group; however, the effect size was small. These changes showed that vitamin C supplementation could decrease the inflammation needed for the breakdown of body fat.

Our study has some limitations. We only presented differences between two groups in elderly women who participated in exercise training for six-weeks. We applied exercises for both groups and had no results for vitamin C supplementation without exercise. Additionally, we did not investigate the peroxidative and antioxidative status within cells and the protein level of the tested genes. Moreover, we only studied thirty-six elderly women from a Polish population. Further research conducted on larger groups of different sexes and ages, with different doses of the vitamin C supplementation, should be carried out.

## 5. Conclusions

An increase in the oxidation of the peroxidation/antioxidation balance was significantly upregulated in the SUPP group, which influenced the differences in response to the applied exercises between groups. These differences were only associated with a decrease in *HSF-1*, *HSPA1A*, and *TNF-α* in the CON group. Depending on the desired or expected outcomes, modulation of *HSF-1* may be positive when it leads to a higher level of *HSPA1A* expression; however, adaptation to exercise was lowered by 1000 mg vitamin C supplementation. Thus, it can be concluded, that the applied dose of vitamin C was too high for elderly women with a normal level of vitamin C in their plasma. In conclusion, the obtained data do not confirm the need for supplementation with 1000 mg of vitamin C in healthy elderly women with no deficiencies identified in vitamin C plasma status and subjected to exercise.

## Figures and Tables

**Figure 1 biomedicines-10-02641-f001:**
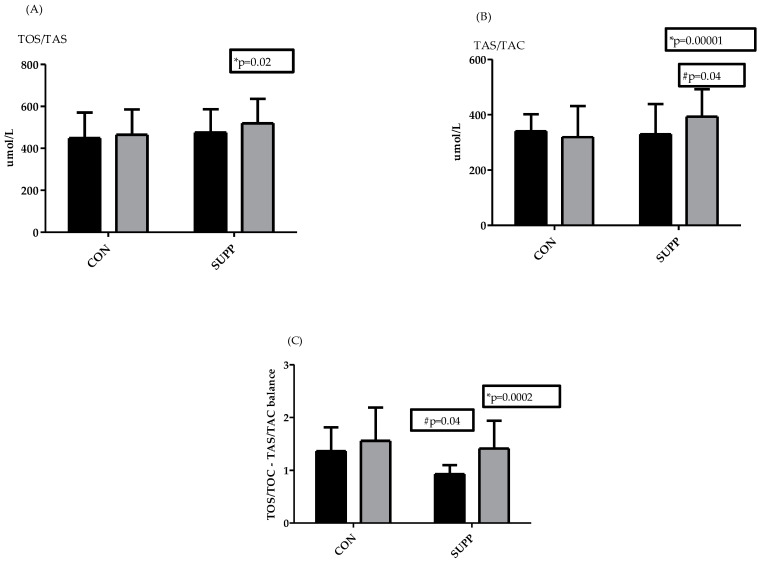
Changes in total oxidative status/capacity (TOS/TOC, (**A**)), total antioxidative status/capacity (TAS/TAC, (**B**)), and pro/antioxidative balance (**C**), before (dark bars) and after 6 weeks (gray bars) in the CON group and SUPP group. * Significant differences between pre/post values (*p* < 0.05); # Significant differences between groups (*p* < 0.05). Reference values: EDTA-plasma: <200 mmol/L low oxidative stress; 200–350 mmol/L moderate oxidative stress; >350 mmol/L high oxidative stress; <280 mmol/L low antioxidative status; 280–320 mmol/L moderate antioxidative status; >320 mmol/L high antioxidative status.

**Figure 2 biomedicines-10-02641-f002:**
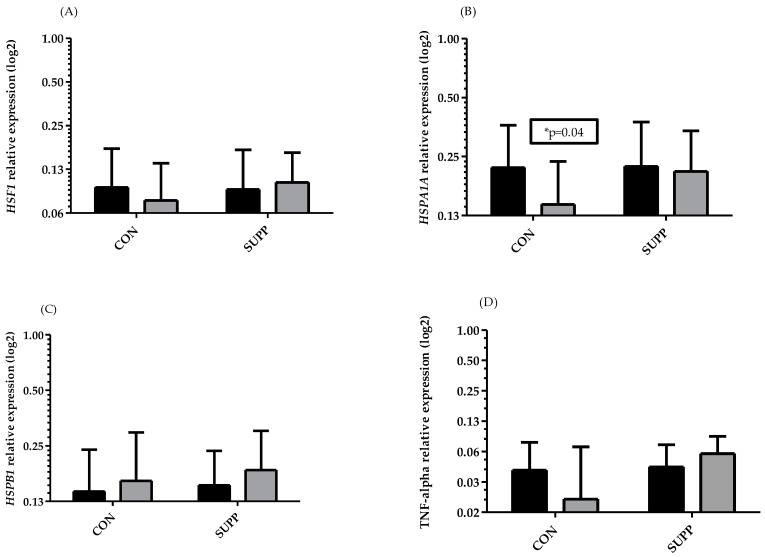
Changes in *HSF-1* (**A**), *HSPA1A* (**B**), *HSPB1* (**C**), and *TNF-α* (**D**) relative expression before (dark bars) and after 6 weeks (gray bars) in the CON group and SUPP group. All data are presented as log 2. * Significant differences between pre/post values (*p* < 0.05).

**Table 1 biomedicines-10-02641-t001:** Changes in body mass, VO_2_ max, and vitamin C concentration in tested groups before and after 6 weeks of supplementation.

Parameter	SUPP Group (n = 18)			dCohen	CON Group (n = 18)			dCohen	2-Way ANOVA
	Baseline (x ± SD)	After (x ± SD)	Change (x ± SD)		Baseline (x ± SD)	After (x ± SD)	Change (x ± SD)		
Body mass (kg)	71.1 ± 9.0	70.4 ± 10.4	−0.7 ± 2.9	0.02	71.3 ± 0.4	68.8 * ± 10.5	−2.4 ± 3.9	0.15	# row factor (*p* < 0.0001)
VO_2_ max. (mL/kg/min)	20.6.± 3.6	21.4 ± 3.9	0.9 ± 1.4	0.19	21.3 ± 3.8	21.4 ± 3.8	0.1 ± 0.75	0.00	# interaction (*p* = 0.0006) # row factor (*p* < 0.0001)
Vit. C concent. (μmol/L)	14.0 ± 3.9	18.0 * ± 7.8	3.9 ± 6.4 ↑ *	0.44	14.5 ± 7.1	14.7 ± 3.9	0.2 ± 5.7	0.08	# row factor (*p* < 0.0001)

* Significant differences between before and after values within group. # Significant differences between groups (ANOVA 2-way). VO_2_ max.—maximal oxygen consumption (mL/kg/min), Vit.C concent.—concentration of Vitamin C in plasma, ↑ significant increase.

## Data Availability

All data are in the possession of the corresponding authors and will be made available upon request.

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
