# Peer review of "Differences in the Pro/Antioxidative Status and Cellular Stress Response in Elderly Women after 6 Weeks of Exercise Training Supported by 1000 mg of Vitamin C Supplementation"

_biomedicines, 2022, doi:10.3390/biomedicines10102641_

Round 1

Reviewer 1 Report

Overall, this ms has some interesting observations. The rationale for the study was clearly stated and the background literature appropriate. However, I found the ms to be incomplete in experimental design, which makes interpretation of the results difficult.

Specific comments:

Title: Specify Vitamin C supplementation.

Abstract: Restate. Blood samples were taken to determine…..

Vit C enhanced “peroxidation” or “prooxidation”?

Vit C increased pro and anti oxidative status? This curious result need further attention in the discussion section. What exactly are the test kit measuring? This needs attention in the text.

The authors state there is a different tendency for expression. This means there was no statistical difference between mean values.

Do the authors use “antioxidant supplementation” and “Vit C supplementation” interchangeably?

2.3 Training: Give a reference for the training schedule. Since all the women trained, we have no effect of exercise on any parameter. THIS IS A MAJOR FLAW IN EXPERIMENTAL DESIGN. For example, if exercise raised a parameter and Vit C mitigated the increase, did Vit C prevent the increase, or lower the value independent of exercise? There are no baseline values.

Table 1: Although the authors maintain no differences in baseline parameters, it appears that the SUPP group had lower plasma Vit C levels.

Fig. 1 A and B: Hard to see how these marginal differences are significant.

Author Response

Response to Reviewer 1:

Dear Reviewer,

Thank You very much for revision our manuscript entitled “Differences in the pro/antioxidative status and cellular stress response in elderly women after 6 weeks of exercise training supported by 1000mg of vitamin C supplementation”.

We considered all comments which helped improve our article. Please find below the answer to each of the comments:

The Reviewer wrote: “Overall, this ms has some interesting observations. The rationale for the study was clearly stated and the background literature appropriate. However, I found the ms to be incomplete in experimental design, which makes interpretation of the results difficult.”.

Response: Thank you very much for this comment. We try to make a revision according to yours suggestion.

Comment 1:” Title: Specify Vitamin C supplementation.”

Response: Please forgive us this inattention. In present version of manuscript, we completed it.

Comment 2: “Abstract: Restate. Blood samples were taken to determine…..”

Response. Thank you very much for this comment. In the abstract we added: Blood samples were taken twice (at baseline and  24 h after whole period of training) in order to determine vitamin C concentration, the total oxidative status/capacity (TOS/TOC), total antioxidant status/capacity (TAS/TAC) and gene expression associated with cellular stress response: encoding heat shock factor (HSF1), heat shock protein 70 (HSPA1A), heat shock protein 27 (HSPB1) and tumor necrosis factor alpha(TNF-a).

Comment 3: “Vit C enhanced “peroxidation” or “prooxidation”?”

Response: Thank you very much for this comment. We change peroxidation to prooxidation. Thank you.

Comment 4: “Vit C increased pro and anti oxidative status? This curious result need further attention in the discussion section. What exactly are the test kit measuring? This needs attention in the text.”

Response: Thank you very much for this comment. In present version we added in method section the description of using KIT and in discussion following fragment: “In PerOX assay, total lipid peroxides level was measured in the sample. PerOx determination was based on the reaction between lipid peroxides and peroxidase with use of photometric detection. ImAnOx determination was based on the reaction of antioxidants present in the sample with added hydrogen peroxide, and the amount of which remaining after the reaction was determined photometrically.”.

Comment 5: “The authors state there is a different tendency for expression. This means there was no statistical difference between mean values.”

Response: Yes, for HSF1, HSPB1 or TNF-a we observed only the tendency, significant differences was noted only for HSPA1A within CON group.

Comment 6: “Do the authors use “antioxidant supplementation” and “Vit C supplementation” interchangeably?”

Response: No, we try to use antioxidant supplementation when not only Vitamin C was supplemented and Vitamin C supplementation only to results obtained when Vitamin C was supplemented alone. We added in introduction: “However, antioxidants supplementation is not only associated with vitamin C, which supplemented alone may results in weaker or different effect on prooxidative/antioxidative balance and then on genes expression [20].”

Comment 7: “2.3 Training: Give a reference for the training schedule. Since all the women trained, we have no effect of exercise on any parameter. THIS IS A MAJOR FLAW IN EXPERIMENTAL DESIGN. For example, if exercise raised a parameter and Vit C mitigated the increase, did Vit C prevent the increase, or lower the value independent of exercise? There are no baseline values.”

Response: Thank you very much for this comment. We added references to applied training. We agree with the Reviewer, that we have only the results for trained women supplemented by 1000 mg Vitamin C and only trained without supplementation. In this case we can discussed only effects of exercise alone and effect of exercise and supplementation. We have not any results for women only supplemented by Vitamin C. We added this in “study limitation”.

Comment 8: “Table 1: Although the authors maintain no differences in baseline parameters, it appears that the SUPP group had lower plasma Vit C levels.”

Response: Thank you very much for this comment. The Vitamin C level in SUPP was 14.0 ± 3.9 at baseline compared to 14.5 ± 7.1 in CON group. This difference was no significant, similarly to differences after intervention. In present version we added the values after 6 weeks of supplementation in both groups. There was high SD in SUPP group and that was the reason for it.

Comment 9: “Fig. 1 A and B: Hard to see how these marginal differences are significant.”.

Response: Thank you very much for this comment. We changed the scale on Y and provided the p values on the figures.

We would like to thank you again for all comments which improved our manuscript.

                                               Kind regards, Małgorzata Żychowska and `Jędrzej Antosiewicz

Reviewer 2 Report

The authors have performed a clinical investigation during 6 weeks in four research centers, to assess the effects of vitamin C supplementation on the oxidative and cellular stress response in elderly women subjected to exercise training.

Several aspects were found in this manuscript:

-    the authors omitted to enter in the title what kind of vitamin was used during the research. Vitamin C is also missing from the keywords, where appears inflammation, a word that has no particular relevance in the context of this study.

-    the introduction and the discussion sections should be more complete, providing supplementary background in the field.

-    the results obtained should be compared with those achieved by other researchers and discussions should be significantly detailed. (see:

·   Dutra MT et al. The effects of strength training combined with vitamin C and E supplementation on skeletal muscle mass and strength: a systematic review and meta-analysis. Journal of Sports Medicine 2020; Article ID 3505209

·   Mason SA et al. Antioxidant supplements and endurance exercise: Current evidence and mechanistic insights. Redox Biol. 2020; 35:101471.

·   Clifford T et al. The effects of vitamin C and E on exercise-induced physiological adaptations: a systematic review and Meta-analysis of randomized controlled trials. Crit Rev Food Sci Nutr. 2020;60(21):3669-3679).

-    in discussion section, the authors need to develop argumentation in depth based on the current understanding and the findings of the results obtained, presenting the potential, the weakness and limitation, and future research direction, among others. Authors should try to explain the theoretical implication as well as the translational application of their research.

- all abbreviations should be expanded in the first appearance. The explanation of the abbreviation should be used only once in the text and should not be repeated, in order to decongest the text and facilitate the understanding of the information transmitted

- some abbreviations are not explained, which creates a lot of confusion and makes the text difficult to read and understand: RNA, mRNA, DNA, cDNA, TUBB, pre/post and other.

- different fonts were used in the text and in the figures;

- missing information (city, country) about the companies producing some devices and the software used in the research;

- the authors should upgrade the references;

- spelling check of the text is mandatory;

- English including grammar, style and syntax, should be improved through the professional help from English Editing Company for Scientific Writings.

Author Response

Response to Reviewer 2:

Dear Reviewer,

Thank You very much for revision our manuscript entitled “Differences in the pro/antioxidative status and cellular stress response in elderly women after 6 weeks of exercise training supported by 1000mg of vitamin C supplementation”.

We considered all comments which helped improve our article. Please find below the answer to each of the comments: “

Comment 1:” the authors omitted to enter in the title what kind of vitamin was used during the research. Vitamin C is also missing from the keywords, where appears inflammation, a word that has no particular relevance in the context of this study.”

Response: Please forgive us this inattention. In present version of manuscript, we completed it. We provided Vitamin C supplementation to keywords and remove inflammation.

Comment 2: “the introduction and the discussion sections should be more complete, providing supplementary background in the field.”

Response: Thank you very much for this comment. In present version of manuscript, we added in Introdaction the following fragment: “The literature data for both aerobic and strength training are ambiguous. Bjørnsen et al. [13]; Close et al [14] and Gomez-Cabrera et al. [15] postulated that high doses of vitamin C may have a negative effect on training adaptation. Moreover, they noted that the age did not have any impact of such resposne. Vitamin C and E supplementation can also alleviate the difficulty of muscle growth by attenuation of pro-inflammatory response and in consequence inhibiting hypertrophy [16]. Therefore, long-duration antioxidant supplementation is not recommended according to training related adaptation of skeletal muscles [16], and “Cliffort et al. [18] in his review indicated that there are not enough data to conclude that vitamin C alone or combined with vitamin E supplementation impairs adaptations in physiological functions, including . molecular level.”  In discussion section we added fragments presented in response 3.

Comment 3: “the results obtained should be compared with those achieved by other researchers and discussions should be significantly detailed”?”

Response: Thank you very much for this comment. In discussion section we added among others: “In this study we assumed, that vitamin C supplementation could increase TAS/TAC and pro/antioxidative balance and thus reduces peroxidation. Genes associated with cellular stress response are easy induced by oxidative stress therefore during supplementation the expression of these genes should be reduced. Current literature associated with genes expression and adaptation to exercises on molecular level are limited. Most data are related to the influence of this supplementation on physiological and biochemical indicators. Thus, some authors postulated no influence on adaptation to training or impediment in adaptation of skeletal muscle mass (anaerobic exercises) or possibilities to endurance training (aerobic exercises) [16,18,34].

Comment 4: “in discussion section, the authors need to develop argumentation in depth based on the current understanding and the findings of the results obtained, presenting the potential, the weakness and limitation, and future research direction, among others. Authors should try to explain the theoretical implication as well as the translational application of their research”.

Response: Thank you very much for this comment. In discussion section we added the following fragment: Our study has some limitations. We present only differences between two groups in elderly women who participated in exercise training for six-weeks. We applied exercises for both group and have no results for vitamin C supplementation without exercise. Additionally, we did not investigate the peroxidative and antioxidative status within cells and proteins level of tested genes. Moreover, we studied only thirty-six elderly women from Polish population. Further research conducted on larger groups of different sexes and ages, with different doses of the vitamin C supplement should be carried out. In conclussions we added: In conclusion, obtained data does not confirm the need for supplementation with 1000 mg of vitamin C in healthy elderly women with no deficiencies identified in vitamin C plasma status and subjected to exercise.

Comment 5: “all abbreviations should be expanded in the first appearance. The explanation of the abbreviation should be used only once in the text and should not be repeated, in order to decongest the text and facilitate the understanding of the information transmitted”

Response: Thank you for this comment. We correct all aabbreviations.

Comment 6: “some abbreviations are not explained, which creates a lot of confusion and makes the text difficult to read and understand: RNA, mRNA, DNA, cDNA, TUBB, pre/post and other.”

Response: Thank you for this comment. We correct this point.

Comment 7: “different fonts were used in the text and in the figures;.”

Response: Thank you very much for this comment. We correct this point.

Comment 8: “missing information (city, country) about the companies producing some devices and the software used in the research.”

Response: Thank you very much for this comment. We try to correct this point.

Comment 9: “the authors should upgrade the references”.

Response: Thank you very much for this comment. We added some authors to references section.

Comment 10: „spelling check of the text is mandatory”

Response: Thank you very much for this comment. We ordered the correction of our manuscript.

Comment 11: “English including grammar, style and syntax, should be improved through the professional help from English Editing Company for Scientific Writings.”

Response: Thank you very much for this comment. We did it and we have the “Certificate”.

We would like to thank you again for all comments which improved our manuscript.

                                                           Kind regards, Małgorzata Żychowska and `jędrzej Antosiewicz

Round 2

Reviewer 1 Report

I still think there are deficits is experimental design.

But you did well in the revision, noting the issues. 

Overall, this is an interesting paper.

Reviewer 2 Report

The authors mostly responded to the comments and suggestions and the manuscript was revised accordingly. I consider it could be accepted for publication in this journal.